# Somatostatin interneurons activated by 5-HT$_{2A}$ receptor suppress slow oscillations in medial entorhinal cortex

**Roberto de Filippo[1,2]\*, Benjamin R Rost[3], Alexander Stumpf[1], Claire Cooper[1], John J Tukker[1,3], Christoph Harms[4,5,6], Prateep Beed[1†], Dietmar Schmitz[1,2,3,6†]\***

[1]Charité–Universitätsmedizin Berlin, corporate member of Freie Universität Berlin, Humboldt-Universität zu Berlin, and Berlin Institute of Health; Neuroscience Research Center, Berlin, Germany; [2]Charité–Universitätsmedizin Berlin, corporate member of Freie Universität Berlin, Humboldt-Universität zu Berlin, and Berlin Institute of Health; Cluster of Excellence NeuroCure, Berlin, Germany; [3]German Centre for Neurodegenerative Diseases (DZNE), Berlin, Germany; [4]Charité–Universitätsmedizin Berlin, corporate member of Freie Universität Berlin, Humboldt-Universität zu Berlin, and Berlin Institute of Health; Department of Experimental Neurology, Berlin, Germany; [5]Charité–Universitätsmedizin Berlin, corporate member of Freie Universität Berlin, Humboldt-Universität zu Berlin, and Berlin Institute of Health; Center for Stroke Research Berlin, Berlin, Germany; [6]Charité–Universitätsmedizin Berlin, corporate member of Freie Universität Berlin, Humboldt-Universität zu Berlin, and Berlin Institute of Health; Einstein Center for Neurosciences Berlin, Berlin, Germany

**\*For correspondence:**
roberto.de-filippo@charite.de
(RF);
dschmitz-office@charite.de (DS)

†These authors contributed equally to this work

**Competing interests:** The authors declare that no competing interests exist.

**Abstract** Serotonin (5-HT) is one of the major neuromodulators present in the mammalian brain and has been shown to play a role in multiple physiological processes. The mechanisms by which 5-HT modulates cortical network activity, however, are not yet fully understood. We investigated the effects of 5-HT on slow oscillations (SOs), a synchronized cortical network activity universally present across species. SOs are observed during anesthesia and are considered to be the default cortical activity pattern. We discovered that (±)3,4-methylenedioxymethamphetamine (MDMA) and fenfluramine, two potent 5-HT releasers, inhibit SOs within the entorhinal cortex (EC) in anesthetized mice. Combining opto- and pharmacogenetic manipulations with in vitro electrophysiological recordings, we uncovered that somatostatin-expressing (Sst) interneurons activated by the 5-HT$_{2A}$ receptor (5-HT$_{2A}$R) play an important role in the suppression of SOs. Since 5-HT$_{2A}$R signaling is involved in the etiology of different psychiatric disorders and mediates the psychological effects of many psychoactive serotonergic drugs, we propose that the newly discovered link between Sst interneurons and 5-HT will contribute to our understanding of these complex topics.

## Introduction

5-HT is one of the most important neuromodulators in the central nervous system. Projections originating from the Raphe nuclei, the brain-stem structure that comprises the majority of 5-HT releasing neurons in the brain, innervate all cortical and sub-cortical areas (*Descarries et al., 2010*). 5-HT levels in the brain are closely linked to the sleep-wake cycle: the activity of serotonergic raphe neurons is increased during wakefulness, decreased during slow-wave sleep (SWS) and virtually silent during REM sleep (*McGinty and Harper, 1976*; *Oikonomou et al., 2019*; *Unger et al., 2020*). Cortical

activity is also influenced by the behavioral state of the animal: SWS is generally associated with 'synchronized' patterns of activity, characterized by low-frequency global fluctuations, whereas active wakefulness and REM sleep feature 'desynchronized' network activity, in which low-frequency fluctuations are absent. The shifting of cortical networks between different patterns of activity is controlled, at least in part, by neuromodulators (*Tukker et al., 2020*; *Lee and Dan, 2012*). For instance, Acetylcholine (ACh) can profoundly alter cortical network activity by inducing desynchronization via activation of Sst interneurons (*Chen et al., 2015*). However, ACh is not solely responsible for suppressing cortical synchronized activity, as lesions of cholinergic neurons are not sufficient to abolish desynchronization (*Kaur et al., 2008*). On the other hand, blocking ACh and 5-HT transmission at the same time causes a continuous 'synchronized' cortical state, even during active behavior, thus suggesting that 5-HT plays an important role in mediating transitions between different network states (*Vanderwolf and Baker, 1986*). In agreement with this line of thought, electrical and optogenetic stimulation of the Raphe nuclei reduce low frequency (1–10 Hz) power in the cortex, implying a reduction in neuronal synchronization at these frequencies (*Puig et al., 2010*; *Grandjean et al., 2019*). Moreover, optogenetic stimulation of serotonergic neurons is sufficient to awaken mice from SWS (*Oikonomou et al., 2019*). These data suggest a relationship between 5-HT levels and patterns of cortical activity (*Lee and Dan, 2012*; *Harris and Thiele, 2011*). The exact mechanism by which 5-HT modulates network activity in the cortex however, is still not fully understood.

Here, we used electrophysiological techniques together with pharmacology, optogenetics, and pharmacogenetics to investigate the effect of 5-HT on slow oscillations (SOs), a network oscillation characterized by synchronized transitions (<1 Hz) between periods of high activity (upstates) and relative quiescence (downstates) (*Steriade et al., 1993*; *Neske, 2015*; *Isomura et al., 2006*). SOs are a global phenomenon observed throughout the cerebral cortex and are considered to be the default emergent activity of cortical networks during SWS and anesthesia (*Neske, 2015*; *Sanchez-Vives et al., 2017*; *Wolansky et al., 2006*). We performed our experiments in the medial entorhinal cortex (mEC), a region where SOs can be recorded both under anesthesia and in acutely prepared cortical slices (*Tahvildari et al., 2012*; *Beed et al., 2020*). Pyramidal neurons located in L3 of mEC provide the excitatory drive underlying each upstate (*Namiki et al., 2013*; *Beed et al., 2020*). Similarly to other mammalian cortical areas, mEC comprises different types of inhibitory GABAergic neurons that can be grouped into three main classes according to immunoreactivity: parvalbumin (PV), somatostatin (Sst), and 5-HT$_3$ (*Miao et al., 2017*; *Rudy et al., 2011*). Most PV neurons target the soma and the spike initiation zone, have low input resistance and minimal spike frequency adaptation. Sst neurons are divided into two groups, one showing features similar to PV interneurons and a second one (i.e. Martinotti cells) that, in contrast, tend to form synapses onto the dendritic trees of their target cells, have high input resistance and show a considerable adaptation. 5-HT$_3$ neurons are usually located in the superficial layers, have high input resistance and mixed adaptation. While these different classes of interneurons are all depolarized during upstates, PV interneurons receive decidedly the strongest excitation (*Tahvildari et al., 2012*; *Neske et al., 2015*). Recurrent excitation and temporal summation of inputs contribute to the transition from downstate to upstate (*Sanchez-Vives et al., 2017*; *Tukker et al., 2020*). This excitation propagates to L5 and differentially entrains L5a and L5b excitatory neurons. L5a neurons do not participate in SOs, whereas L5b neurons are steadily synchronized. Both intrinsic and synaptic mechanisms have been implicated in upstate termination (*Neske, 2015*; *Tukker et al., 2020*). Activity-dependent K$^+$ channels decrease excitability of neurons over time causing a generalized reduction of facilitation (*Neske, 2015*; *Harris and Thiele, 2011*). At the same time, blockage of GABA$_B$ receptors significantly extends upstate duration (*Craig et al., 2013*) and inhibitory drive has been observed to increase during upstate termination (*Lemieux et al., 2015*). Sst interneurons, and in particular Martinotti cells, characterized by strongly facilitating synapses (*Beierlein et al., 2003*; *Gibson et al., 1999*) have been proposed as an important source of inhibition in the termination of upstates (*Melamed et al., 2008*; *Krishnamurthy et al., 2012*).

Our results show that (±)3,4-methylenedioxymethamphetamine (MDMA) and fenfluramine (Fen), two substitute amphetamines that induce robust 5-HT release, suppress SOs in anesthetized mice and, concurrently, activate a small group of neurons characterized by an intermediate waveform shape. Using cortical slices in vitro, we demonstrate that 5-HT inhibits SOs in the mEC and that Sst interneurons, activated by 5-HT$_{2A}$R, are involved in the modulatory effect of 5-HT. While previous studies have shown that parvalbumin (PV) interneurons are excited by 5-HT$_{2A}$R (*Puig et al., 2010*;

*Athilingam et al., 2017*), our results identify cortical Sst interneurons as novel targets of the 5-HT neuromodulatory system via 5-HT$_{2A}$R.

## Results

### Pharmacological release of 5-HT inhibits SOs in anesthetized mice

We investigated the effect of 5-HT on network activity in anesthetized mice using multisite silicon microelectrodes placed in the mEC L3, a region located in the medial temporal lobe interconnected to a variety of cortical and subcortical areas, including the Raphe nuclei (*Figure 1A*; *van Strien et al., 2009*; *Muzerelle et al., 2016*). Under urethane anesthesia, EC, like the rest of the cortex, displays SOs (*Figure 1B*). As expected, we found that upstates were present synchronously in the local field potential (LFP) of all the recording channels (*Figure 1—figure supplement 1*), and that every upstate coincided with large increases in population spiking activity (*Figure 1B*). 5-HT does not cross the blood brain barrier (*Hardebo and Owman, 1980*), therefore, to understand the effect of 5-HT on SOs we used MDMA, a potent presynaptic 5-HT releaser and popular recreational drug (*Green et al., 2003*) that has shown promising results in the treatment of post-traumatic stress disorder (PTSD) (*Inserra et al., 2021*).

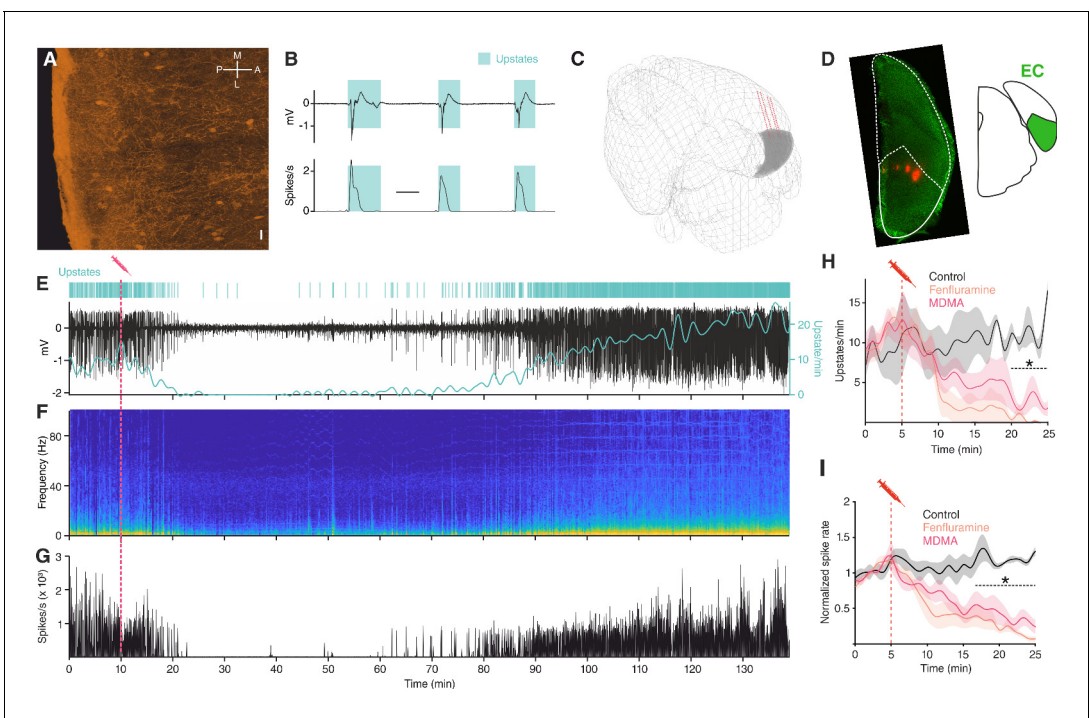

**Figure 1.** MDMA/Fen inhibit SOs in anesthetized mice. (**A**) Immunocytochemical analysis of an ePet-YFP mouse showing serotonergic fibers in medial entorhinal cortex, horizontal slice (M = medial, L = lateral, P = posterior, A = anterior). Scale bar: 20 μm. (**B**) LFP (top) and instantaneous population activity (bottom) of a representative in vivo recording during SOs (spikes/s units in thousands), cyan rectangles represent detected upstates. (**C**) 3D visualization of the microelectrode location of the recording shown in E. EC represented in gray. (**D**) Left: microelectrode tracks (red) of the recording shown in (**E**). Right: EC position represented in green. (**E**) Top: Cyan lines represent detected upstates. Bottom: LFP (black) and average upstate incidence per minute (cyan). Pink dotted line represents MDMA application time. (**F**) Fourier transformation and (**G**) instantaneous population activity for the recording shown in E. (**H**) Mean upstate incidence after saline (control), Fen or MDMA application (control: n = 5, Fen: n = 6, MDMA: n = 7; p<10$^{-4}$, unpaired t test with Holm-Šidák correction). (**I**) Mean normalized spike rate after saline (control), Fen or MDMA application (control: n = 5, Fen: n = 6, MDMA: n = 7; p<10$^{-4}$, unpaired t test with Holm-Šidák correction).

The online version of this article includes the following figure supplement(s) for figure 1:

**Figure supplement 1.** In vivo upstate spatial features.

**Figure supplement 2.** LFP power analysis for saline and MDMA/Fen injection.

**Figure supplement 3.** In vivo upstate metrics for saline and MDMA/Fen injection.

Intraperitoneal injections of MDMA (1.25 mg/kg) caused a strong suppression of upstate incidence (*Figure 1E–H*), a decrease in power of low frequencies (1–20 Hz) (*Figure 1F*), and a reduction of population spiking activity (*Figure 1G–I*). In addition to 5-HT, MDMA has been shown to cause the release of other neuromodulators, such as dopamine and noradrenaline (NE), although to a much lesser extent (*Green et al., 2003*). To test whether the effect of MDMA was mediated specifically by 5-HT, we repeated the experiment using Fen (5 mg/kg), a more selective 5-HT releaser (*Rothman and Baumann, 2002*; *Heifets et al., 2019*; *Baumann et al., 2000*). Intraperitoneal injection of Fen had a comparably strong suppressive effect on both the occurrence of upstates and population spiking activity (*Figure 1H–I*). Given the similarity of the observed effects, we grouped the results of Fen and MDMA application and found that both of these drugs significantly decreased LFP power in the delta and theta frequency ranges (*Figure 1—figure supplement 2*). Furthermore, the duration and area of upstates were also significantly reduced (*Figure 1—figure supplement 3*). These results point to a likely involvement of 5-HT in modulating ongoing oscillatory activity and suppressing low-frequency fluctuations.

## MDMA and Fen activate a subgroup of cortical neurons

In addition to the LFP signal, we recorded the activity of 355 single units within the EC. Because of the similar effect of MDMA and Fen on spike rates (*Figure 1I*), we pooled all units recorded in both

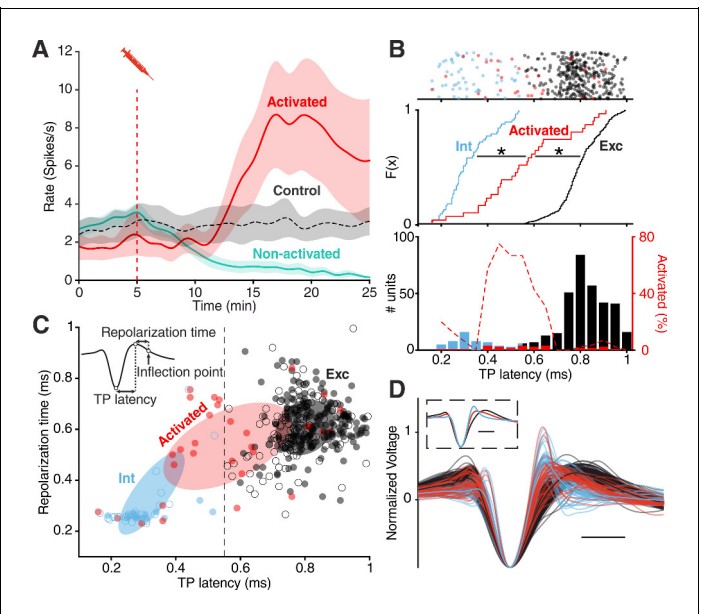

**Figure 2.** Divergent unit responses to MDMA/Fen application. (**A**) Spike rate of the activated units versus all the other units during MDMA/Fen application (activated: n = 31, Non-activated: n = 324). (**B**) Top: TP latencies color-coded by group. Middle: cumulative distribution of TP latencies (Kolmogorov-Smirnov test, p Activated vs Int <10–4, p Activated vs Exc <10–4). Bottom: bar plot representing probability distribution of TP latencies, on the right y axis dashed line representing the percentage of 'activated' units per TP latency bin. (**C**) Distribution of units according to trough-to-peak (TP) latencies and repolarization time. Units were classified as putative interneurons (Int, blue) and putative excitatory neurons (Exc, dark gray) according to a threshold at 0.55 ms; activated units (red) could belong to either group but were mostly intermediate as shown by the covariance (2 STD) of each group (Ellipses). Units recorded during control experiments are represented by empty circles. (**D**) Waveforms of recorded units (n = 355). Units were divided into 'putative excitatory' (black) and 'putative inhibitory' (blue) neurons according to TP latencies. Units activated by either MDMA or Fen application are represented in red. Inset shows the average waveform for each group. Scale bars: 0.5 ms.

The online version of this article includes the following source data and figure supplement(s) for figure 2:

**Source data 1.** Source data for *Figure 2A*.

**Figure supplement 1.** Cross-correlogram (CCG) based connectivity analysis.

**Figure supplement 2.** TP latency density distributions.

types of experiments. We found that drug injections differentially affected spiking rates (*Figure 2A*) of recorded units: while spiking decreased in most units ('non-activated'), a small group of units ('activated') responded in the opposite fashion (n = 31/355, 8.7%).

We then calculated the trough-to-peak (TP) latency of the spike waveforms, which has been consistently used as a metric to classify units. In accordance with previous studies (*Senzai et al., 2019*; *Roux et al., 2014*), we found a clear bimodal distribution of TP latencies distinguishing putative excitatory (Exc) and fast-spiking inhibitory (Int) groups. Analysis of cross-correlograms confirmed the inhibitory nature of a subset of putative FS units (*Figure 2—figure supplement 1*; *Barthó et al.,*

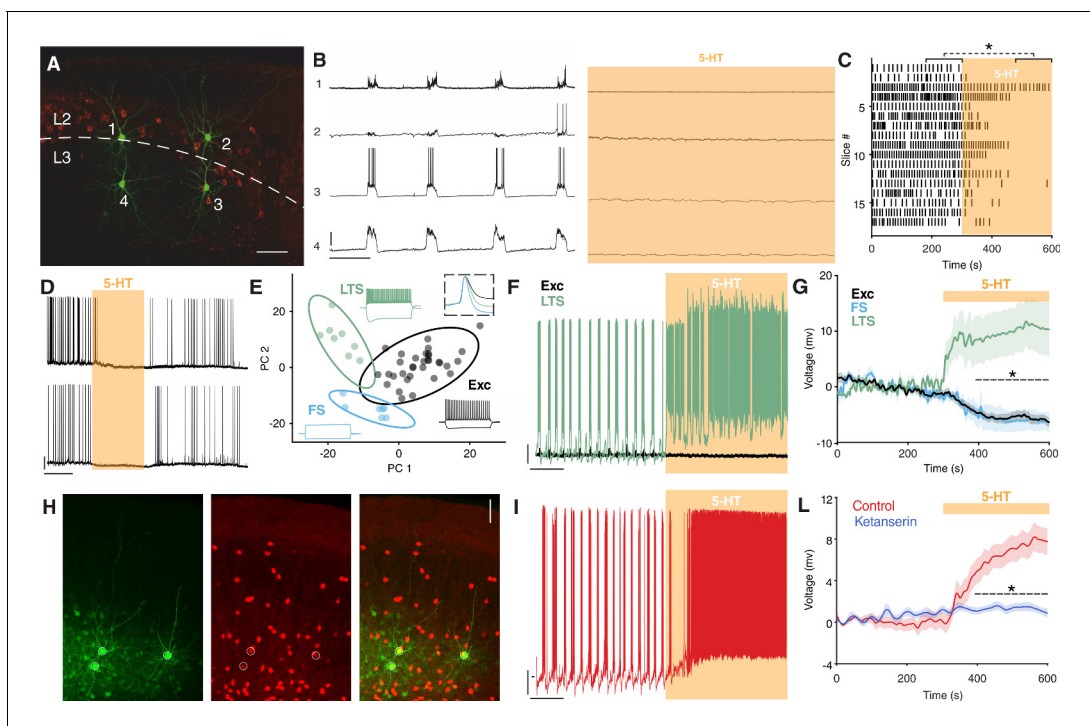

**Figure 3.** 5-HT suppresses SOs and activates Sst interneurons. (**A**) Biocytin staining of four simultaneously recorded cells shown in (**B**) WFS1 expression (in red) delimits L2/3 border. (**B**) Intracellular recordings showing synchronous upstate events in four simultaneously recorded cells before (left) and after (right) 5-HT application. Scale bars: 1: 7.5 mV, 2: 25 mV, 3: 25 mV, 4: 10 mV; 10 s. (**C**) Upstate raster plot before and after 5-HT application, orange box represents 5-HT application (n = 17, $p<10^{-4}$, Wilcoxon signed rank test). (**D**) Representative recording showing the temporary inhibitory effect of 5-HT on SOs in two simultaneously recorded cells. Scale bars: 5 min, 20 mV. (**E**) PCA projection plot of all the cells recorded (n = 48). Cells are color-coded according to group identity: Exc (black), FS (light blue), or LTS (green). Typical voltage responses to current injection (−150 and +250 pA) are plotted for each group. Inset shows the average spike waveform for each group. (**F**) Representative recording of an excitatory (black) and a low-threshold (green) neuron simultaneously recorded during 5-HT application. Scale bars: 10 mV, 30 s. (**G**) Average change of RP before and after 5-HT application, across excitatory, fast-spiking and low-threshold neurons (Exc: n = 34, FS: n = 6; LTS: n = 9; $p<10^{-4}$, unpaired t test with Holm-Šidák correction). (**H**) Biocytin staining of cells recorded in Sst-tdTomato mouse. Biocytin in green, tdTomato in red. Scale bar: 50 μm. (**I**) Representative recording of a Sst interneuron during 5-HT application. Scale bars: 10 mV, 30 s. (**L**) Average RP of Sst interneurons during 5-HT (red) and ketanserin + 5-HT (blue) application, orange bar represents 5-HT (5-HT: n = 19, ketanserin + 5-HT: n = 22).

The online version of this article includes the following source data and figure supplement(s) for figure 3:

**Source data 1.** Source data for *Figure 3G*.
**Source data 2.** Source data for *Figure 3L*.
**Figure supplement 1.** Effect of Fen on SOs in vitro.
**Figure supplement 2.** 5-HT suppresses evoked upstates in vitro.
**Figure supplement 3.** 5-HT$_{2A}$Rs are involved in 5-HT mediated SOs suppression in vitro.
**Figure supplement 4.** 5-HT$_3$ receptor is not involved in 5-HT mediated SOs suppression.
**Figure supplement 5.** In vitro upstates metrics during baseline and ketanserin + 5-HT application.
**Figure supplement 6.** LTS neurons are depolarized by 5-HT.
**Figure supplement 7.** Excitatory, fast-spiking, and LTS cells have unique sets of electrophysiological features.
**Figure supplement 8.** Classification of cells recorded in Sst-tdTomato mice.
**Figure supplement 9.** Spatial localization of 5-HT$_{2A}$R positive cells in EC.

*2004*). The cumulative distribution of TP latencies of 'activated' units was significantly different from both Exc and Int groups (*Figure 2B*). Specifically, the average TP latency of 'activated' units was situated in between the Int and Exc groups (*Figure 2B–D*, *Figure 2—figure supplement 2*), possibly suggesting a non-FS interneuron identity (*Trainito et al., 2019*; *Kvitsiani et al., 2013*). Notably, the intermediate TP latency of 'activated' units is not the result of an equal distribution between high and low values. Units with intermediate TP latency (between 0.4 and 0.6 ms) are unique in showing a 40–80% likelihood of being 'activated' by drug application (*Figure 2B*).

## 5-HT suppresses SOs and activates Sst interneurons via 5-HT$_{2A}$R in vitro

To understand the mechanism underlying the suppression of SOs by MDMA/Fen, we performed in vitro experiments combining electrophysiology and pharmacology. First, we recorded simultaneously from up to four neurons in the superficial layers of the EC (*Figure 3A*). Brain slices were perfused with an extracellular solution containing $Mg^{2+}$ and $Ca^{2+}$ in concentrations similar to physiological conditions. With this method we could reliably detect SOs reminiscent of the in vivo network activity (*Tahvildari et al., 2012*). Release of 5-HT in vitro, induced by Fen application (200 µM), caused a suppression of SOs similar to what we observed in vivo (*Figure 3—figure supplement 1*). Likewise, application of low concentrations of 5-HT (5 µM) resulted in the suppression of SOs (*Figure 3B,C*). This effect was highly consistent across different slices and was readily reversible (*Figure 3D*). Similarly to spontaneous upstates, electrically evoked upstates (*Neske et al., 2015*) were also suppressed by 5-HT (*Figure 3—figure supplement 2*). Increasing the stimulation intensity did not rescue upstate generation, indicating that a lack of excitation alone cannot explain the suppressive effect of 5-HT on SO. In accordance with these results, we decided to focus solely on 5-HT, however, we acknowledge that other neurotransmitters might be involved in the effect of MDMA/Fen we observed in vivo.

Suppression of activity can have either an intrinsic or synaptic origin (*Turrigiano, 2011*). A substantial subset of EC excitatory neurons is known to express 5-HT$_{1A}$ receptor (5-HT$_{1A}$R) and hyperpolarize upon 5-HT application via activation of G protein-coupled inwardly rectifying potassium (GIRK) channels (*Schmitz et al., 1998*; *Chalmers and Watson, 1991*). The suppression of SOs by 5-HT, however, was not influenced by blocking 5-HT$_{1A}$R (WAY 100635, 100 nM) (*Figure 3—figure supplement 3A,E*). Blocking 5-HT$_3$R (tropisetron, 1 µM), the receptor that characterizes one of the three main groups of interneurons, also did not have any impact on SOs suppression induced by 5-HT. Similarly, application of the 5-HT$_3$R agonist m-CPBG (50 µM) did not have any effect on SOs (*Figure 3—figure supplement 4*). In contrast, blocking 5-HT$_{2A}$R with the selective antagonist ketanserin (1 µM) (*Preller et al., 2018*) strongly reduced the suppression power of 5-HT on SOs from 95 ± 4% to 57 ± 10.1% (*Figure 4—figure supplement 1B,E*). The remaining suppression can be possibly explained by the activation of 5-HT$_{1A}$R on excitatory cells, as is reflected by the reduced spiking activity of putative excitatory cells (*Figure 3—figure supplement 5*).

Selective activation of 5-HT$_{2A}$R by α-methyl-5-HT (5 µM) mimicked the suppression of SOs observed after 5-HT wash-in (*Figure 3—figure supplement 3C,E*). Together, these results point to the importance of 5-HT$_{2A}$R in the suppression of SOs. 5-HT$_{2A}$R activation is known to cause an increase in intracellular calcium and consequent depolarization of the resting potential (RP) (*Nichols and Nichols, 2008*). Accordingly, after 5-HT application, we found that a small group of neurons was depolarized (n = 6/48, 12.5%) (*Figure 3—figure supplement 6B–C*). Using a soft clustering approach with six electrophysiological parameters (see 'Materials and methods') we divided the recorded cells into three groups: Excitatory (Exc), fast-spiking (FS), and low-threshold spiking (LTS). Strikingly, the cells excited by 5-HT belonged exclusively to the LTS group (*Figure 3G*, *Figure 3—figure supplement 6*). A substantial proportion of LTS neurons express Sst (*Tremblay et al., 2016*; *Gibson et al., 1999*); therefore, we performed targeted patch-clamp recordings using a mouse line expressing tdTomato specifically in Sst-expressing interneurons (*Figure 3H*). A subgroup of Sst interneurons are FS cells (*Urban-Ciecko et al., 2018*), only LTS Sst interneurons were considered in the following analysis. Sst interneurons depolarized upon 5-HT application (n = 19, ΔRP: 7.5 ± 1.23 mV) (*Figure 3I–L*) and in some cases continued to spike while SOs were suppressed (n = 8/17, 47.05%, mean $_{\text{spiking rate}}$ = 3.03 ± 0.39 spikes/s). This effect was blocked by ketanserin (n = 22) (*Figure 3L*). We confirmed the presence of 5-HT$_{2A}$R in Sst interneurons using immunohistochemistry in mice expressing EGFP under the 5-HT$_{2A}$R promoter. In accordance with a previous study we found the majority of 5-HT$_{2A}$R positive cells in the deep layers of EC, with a peak

in L6 (*Weber and Andrade, 2010*). We observed that $11.8 \pm 2.9\%$ of the 5-HT$_{2A}$R positive cells in EC colocalized with Sst (n = 7 mice) (*Figure 3—figure supplement 9*). These results suggest that Sst interneurons may provide the synaptic inhibition required for the suppression of SOs.

## Sst interneurons mediate the suppression of SOs by 5-HT in vitro

To evaluate the contribution of Sst interneurons to the 5-HT-mediated silencing of SOs we used an opto- and pharmacogenetic approach. First, we transgenically expressed channelrhodopsin-2 (ChR2) in Sst interneurons (*Figure 4A*). Light-stimulation of ChR2- expressing Sst interneurons in vitro suppressed SOs consistently (*Figure 4D–F*). Expectedly, upstate-associated spiking was also diminished (*Figure 4C,E,F*). At the end of the light stimulation, spontaneous upstates immediately reoccurred (*Figure 4G–H*), in line with a critical role of Sst interneurons in the modulation of SOs

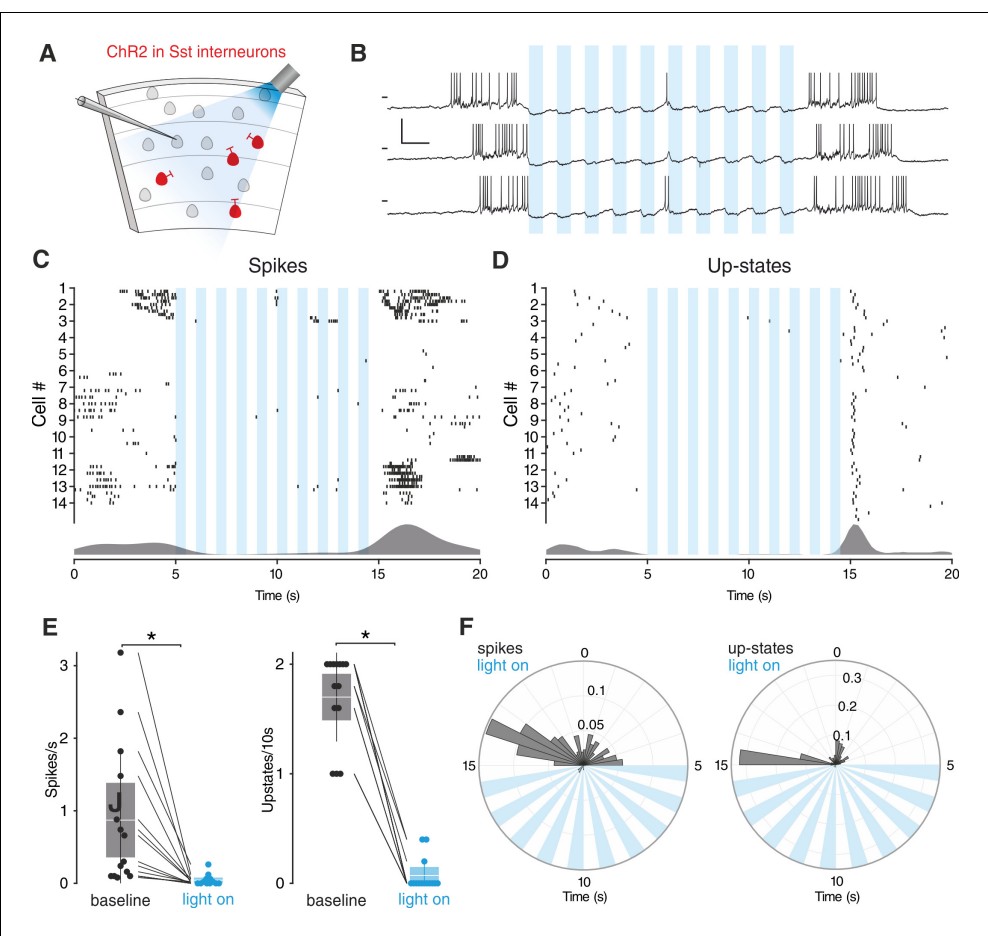

**Figure 4.** Sst interneurons activation suppresses SOs. (**A**) Experimental protocol: Sst interneurons expressing ChR2 are activated by light during intracellular recording of L3 neurons in EC. (**B**) Representative recordings from a L3 neuron during Sst interneuron activation. Scale bars: 10 mV, 0.5 s. (**C**) Spikes raster (top) and density plot (bottom) during light stimulation. (**D**) Upstate raster (top) and density plot (bottom) during light stimulation. (**E**) Left: spike frequency during baseline light stimulation (n = 14; p<0.001, Wilcoxon signed rank test). Right: upstate incidence during baseline and light stimulation (n = 14; p<0.001, Wilcoxon signed rank test). Patches represent 95% confidence interval, lines represent standard deviation. (**F**) Left: spike probability polar plot during Sst interneurons light activation. Right: upstate probability polar plot during Sst interneurons light activation. Note the absence of both spiking activity and upstates during Sst interneurons activation.

The online version of this article includes the following source data and figure supplement(s) for figure 4:

**Source data 1.** Source data for *Figure 4E*.

**Figure supplement 1.** Vector construction and RMCE for the generation of a transgenic mouse line with Cre-conditional hM4Di expression.

(*Fanselow et al., 2008*; *Funk et al., 2017*; *Niethard et al., 2018*). We acknowledge that activation of PV interneurons can cause a similar suppression (*Zucca et al., 2017*). While this experiment establishes the ability of Sst interneurons to suppress SOs, it does not causally link Sst interneuron activation to the suppression of SOs induced by 5-HT. Therefore, we generated a transgenic mouse line carrying a Cre-conditional expression cassette of the pharmacogenetic silencer hM4Di (*Figure 4—figure supplement 1*; *Armbruster et al., 2007*). Homozygous Cre-conditional hM4Di transgenic mice and Sst-Cre mice were bred to obtain heterozygous Sst-Cre/hM4Di offspring, which allow specific inhibition of Sst interneuron activity using Clozapine-N-Oxide (CNO). Following application of 5-HT we observed a strong reduction of upstate incidence, which was partially restored by subsequent application of CNO (*Figure 5A–B*).

Activation of 5-HT$_{1A}$R on excitatory cells by 5-HT and the resulting decreased network excitation drive might account for the fact that the incidence of upstates did not completely return to the baseline level upon CNO wash-in. Additionally, in further experiments in which CNO was applied before 5-HT, the significant reduction in upstate incidence typically seen after 5-HT wash-in was not observed (*Figure 5—figure supplement 1*). CNO did not show any significant effect in both wild-type littermates and hM4Di-PV mice (*Figure 5—figure supplement 2*), indicating that the results observed are due to its specific effect on the activity of Sst interneurons. In summary, while activation of Sst interneurons either via 5-HT or directly by ChR2 suppresses SOs, the pharmacogenetic inactivation of Sst interneurons weakens the effect of 5-HT on SOs.

## Discussion

In this study, we show that the substitute amphetamines MDMA and Fen suppress default cortical network oscillations in vivo in the mEC. Furthermore, using an opto- and pharmacogenetic approach in vitro, we demonstrate that Sst interneurons, activated by 5-HT$_{2A}$R, contribute to this suppression.

Organization of cortical activity is brain state-dependent, ranging continuously from 'synchronized' to 'desynchronized' states (*Harris and Thiele, 2011*). SOs are on one end of this continuum, representing the prototypical synchronized state. Our results, in line with previous studies (*Puig et al., 2010*; *Grandjean et al., 2019*), show that 5-HT can suppress synchronized cortical activity; in addition, we identify Sst interneurons as contributing to this suppression.

Sst interneurons have been previously proposed to provide the inhibition necessary for the termination of upstates, due to their strongly facilitating synapses (*Krishnamurthy et al., 2012*; *Melamed et al., 2008*). Additionally, increased inhibition, as shown both in computational models and experimental data, can counteract temporal summation of inputs and reduce correlation due to tracking of shared inputs between inhibitory and excitatory populations (*Sippy and Yuste, 2013*; *Renart et al., 2010*; *Stringer et al., 2016*). In the case of Sst interneurons, this would especially impact summation in distal compartments of pyramidal cells, due to the axonic targeting of Sst cells onto pyramids. Sst interneurons are specifically known to be sufficient to cause desynchronization in V1 (*Chen et al., 2015*). While it is well known that Sst interneurons are potently excited by ACh in various cortical areas (*Chen et al., 2015*; *Obermayer et al., 2018*; *Fanselow et al., 2008*), including mEC (*Desikan et al., 2018*) and NE in frontal cortex (*Kawaguchi and Kubota, 1998*), our work identifies them as a novel target of 5-HT regulation via 5-HT$_{2A}$R. Recently, it has

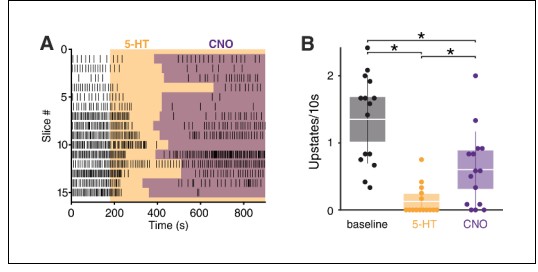

**Figure 5.** Sst interneurons mediate the effect of 5-HT on SOs. ( **A**) Upstate raster plot during 5-HT and subsequent CNO application. Orange box represents 5-HT, purple boxes represent CNO. Note the appearance of upstates after CNO application. (**B**) Upstate incidence during 5-HT and 5-HT+CNO application (n = 15; p (baseline vs 5-HT)<10–4, p (baseline vs CNO)=0.0482, p5-HT vs CNO = 0.0405, Kruskal-Wallis test). Patches represent 95% confidence intervals, lines represent standard deviation.

The online version of this article includes the following source data and figure supplement(s) for figure 5:

**Source data 1.** Source data for *Figure 5B*.

**Figure supplement 1.** CNO application prevents 5-HT mediated upstates suppression.

**Figure supplement 2.** CNO application in wild-type littermates and PV-hM4Di mice.

been proposed that inhibitory interneurons play a key role in mediating the effect of ACh and NE on cortical state transitions (*Cardin, 2019*). Our results add a new level of complexity to this picture.

The excitation of Sst interneurons by 5-HT possibly contributes to the net inhibitory effect of 5-HT release observed in many cortical areas (*Grandjean et al., 2019*; *Seillier et al., 2017*; *Azimi et al., 2020*), and could explain why the inhibition strength is linearly correlated to 5-HT$_{2A}$R expression (*Grandjean et al., 2019*). Giving further support to this idea, Sst interneurons in the somatosensory cortex show increased cFos levels following 5-HT$_{2A}$R activation (*Martin and Nichols, 2016*). Previous studies have reported either 5-HT$_{2A}$R-dependent inhibition or 5-HT$_{2A}$R-dependent activation of interneurons in the prefrontal cortex (PFC) (*Abi-Saab et al., 1999*; *Ashby et al., 1990*; *Athilingam et al., 2017*), piriform cortex (*Marek and Aghajanian, 1994*; *Sheldon and Aghajanian, 1990*), cingulate cortex (*Zhou and Hablitz, 1999*), cochlear nucleus (*Tang and Trussell, 2017*), amygdala (*Sengupta et al., 2017*), olfactory bulb (*Petzold et al., 2009*; *Hardy et al., 2005*), visual cortex (*Michaiel et al., 2019*; *Azimi et al., 2020*), and hippocampus (*Wyskiel and Andrade, 2016*). However, none of these studies identified interneurons using molecular markers, so we do not exclude that different interneuron classes in other cortical areas might mediate the inhibitory downstream effects of 5-HT$_{2A}$R. For example, in the PFC a subgroup of PV interneurons has been reported to be activated by this receptor (*Athilingam et al., 2017*; *Puig et al., 2010*). Furthermore, we know that PV neurons can reliably induce up to down state transitions (*Zucca et al., 2017*). 5-HT modulation is also involved in gain regulation. In the olfactory cortex, 5-HT has a selective subtractive effect on stimulus evoked firing (*Lottem et al., 2016*), and a recent study has shown in the visual cortex that the reduced gain of evoked responses is dependent on 5-HT$_{2A}$R activation (*Azimi et al., 2020*). Intriguingly, Sst interneurons have been shown to regulate subtractive inhibition (*Sturgill and Isaacson, 2015*; *Wilson et al., 2012*).

5-HT levels in the brain, similarly to other neuromodulators, are synchronized to the sleep wake cycle, with higher levels present during the waking state (*Oikonomou et al., 2019*). How does this notion relate to activation of Sst interneurons by 5-HT$_{2A}$R? A limitation of our study is the absence of data in naturalistic conditions, with anesthesia potentially being a confounding factor (*Adesnik et al., 2012*). We can nonetheless speculate that Sst interneurons should be more active during states with higher 5-HT levels (wake>SWS>REM). A previous study directly measuring the activity of various neuronal classes across different states seems to support this idea. Sst interneurons in the dorsal cortical surface display their highest activity during waking states, lower activity during SWS and lowest activity during REM (*Niethard et al., 2016*). Following pharmacological release of 5-HT in vivo, we observed increased spiking activity in a subgroup of neurons whose waveform features are compatible with Sst interneurons. Our in vitro data using 5-HT at a concentration commonly used in the field (*Gorinski et al., 2019*; *Wang et al., 2016*; *Huang et al., 2009*) shows that Sst interneurons can be activated by 5-HT$_{2A}$R, however, recordings during natural sleep and wake conditions in combination with optotagging will be needed to demonstrate conclusively that Sst interneurons are activated by 5-HT in physiological conditions in vivo. Potential differences between physiological and pharmacological 5-HT release are especially of interest considering the reported efficacy of MDMA in the treatment of PTSD and its recently approved status as breakthrough therapy (*Inserra et al., 2021*; *Mithoefer et al., 2019*).

While our data relative to MDMA application supports a role of 5-HT in modulating SOs in vivo we cannot exclude that other neuromodulators might play a role. For example, it is known that dopamine can suppress SOs in vitro (*Mayne et al., 2013*). Fen, however, has been reported to selectively increase 5-HT concentration in the brain (*Rothman and Baumann, 2002*) and has been used previously to specifically disentangle the effect of MDMA on different neuromodulatory systems (*Heifets et al., 2019*). This, together with our in vitro results, suggests that 5-HT is most likely mediating the effect induced by MDMA/Fen.

Besides its involvement in various physiological brain processes, 5-HT is also associated with the etiology of various psychiatric disorders, as are Sst interneurons (*Pantazopoulos et al., 2017*; *Lin and Sibille, 2015*). Furthermore, 5-HT is linked to the psychological effect of many psychotropic drugs; specifically, 5-HT$_{2A}$R activation has been reported to be essential for the psychological effects induced by various psychedelics (*Nichols, 2016*), and in the case of MDMA, has been linked to perceptual and emotional alterations (*Liechti et al., 2000*; *Kuypers et al., 2018*). Broadband reduction in oscillatory power, triggered by 5-HT$_{2A}$R, seems to be linked to the subjective effect of serotonergic drugs (*Carhart-Harris et al., 2016*; *Carhart-Harris and Friston, 2019*) and has been consistently

observed in humans and rodents following administration of MDMA (*Frei et al., 2001*; *Lansbergen et al., 2011*) or various other 5-HT$_{2A}$R agonists (*Kometer et al., 2015*; *Muthukumaraswamy et al., 2013*; *Carhart-Harris et al., 2016*; *Wood et al., 2012*). The link between 5-HT$_{2A}$R and perception is further supported by the fact that several routinely used antipsychotic drugs are potent 5-HT$_{2A}$R antagonists (*Marek et al., 2003*; *Meltzer, 1999*). Although the most recent attempts to explain the psychological effects of 5-HT$_{2A}$R activation focus on the increased spiking of cortical pyramidal neurons in the deep layers (*Carhart-Harris and Friston, 2019*; *Nichols, 2016*), our study suggests that Sst interneurons may also play a role. Sst interneurons, in contrast to PV interneurons, form synapses on the dendrites of their target cell (*Tremblay et al., 2016*). A wealth of evidence suggests that active dendritic processing in cortical pyramidal neurons has a critical influence on sensory perception (*Takahashi et al., 2016*; *Murayama et al., 2009*; *Smith et al., 2013*; *Ranganathan et al., 2018*) and, in accordance with their unique anatomical properties, Sst interneurons strongly influence dendritic computations and directly modulate perceptual thresholds (*Takahashi et al., 2016*).

We propose that the novel link between 5-HT$_{2A}$R and Sst interneurons might help elucidate the mechanisms underlying a host of psychiatric disorders and contribute to our understanding of how serotonergic drugs exert their psychological effects.

## Materials and methods

All experiments were conducted according to regulations of the Landesamt für Gesundheit und Soziales (Berlin [T 0100/03], Berlin [G0298/18]) and the European legislation (European Directive 2010/63/EU).

### Animals

Data for the in vivo part of the study was collected from C57Bl/6 mice (aged 6–10 weeks). Data for the in vitro part was collected from C57Bl/6 (P10-P17), Sst-tdTomato (P10-P30), Sst-Chr2-EYFP (P10-P16), hM4Di-Sst (P10-P15), hM4Di-Sst (+/-) (P10-P15) and hM4Di-PV (P10-P15) mice. Immunostainings to localize 5-HT$_{2A}$R were performed on 5-HT$_{2A}$R-EGFP mice (P20-P90) and immunostainings to localize 5-HT fibers were performed on an ePet-YFP mouse (P35). Sst-Cre mice (RRID:IMSR_JAX: 013044) have Cre recombinase targeted to the *Sst* locus; they were obtained from Jackson Laboratory (ME, USA). PV-Cre mice (RRID:IMSR_JAX:008069) have Cre recombinase targeted to the *Pvalb* locus; they were obtained from Charles River Laboratory (MA, USA). tdTomato mice (Ai9, RRID: IMSR_JAX:013044) were obtained from Jackson Laboratory. Chr2-EYFP mice (Ai32, RRID:IMSR_JAX: 012569) were obtained from Jackson Laboratory. ePet-Cre mice (RRID:IMSR_JAX:012712) have Cre recombinase targeted to the *Fev* locus; they were obtained from Jackson Laboratory. 5-HT$_{2A}$R-EGFP mice (RRID:MMRRC_010915-UCD) express EGFP reporter protein under the control of the *Htr2a* gene, they were obtained from the Mutant Mouse Resource and Research Centers (MMRRC, CA, USA). Generation of hM4Di mice is described in the paragraph 'Generation of Cre-conditional hM4Di mice'. The animals were housed in a 12:12 hr light-dark cycle in singularly ventilated cages with ad libitum access to food and water. SOs in vitro recordings were performed on P12-P16 mice.

### Generation of Cre-conditional hM4Di mice

We produced a transgenic mouse line carrying a Cre-conditional hM4Di expression cassette in the Rosa26 locus. The transgene construct was inserted by recombination-mediated cassette exchange (RMCE). RMCE relies on recombination events between attB and attP recognition sites of the RMCE plasmid and genetically modified acceptor embryonic stem (ES) cells, mediated by the integrase of phage phiC31 (*Hitz et al., 2007*). The RMCE construct is thereby shuttled into the Rosa26 locus of the ES cells, along with a Neomycin resistance cassette (*Figure 1—figure supplement 2A*). The acceptor cell line IDG3.2-R26.10–3 (I3) was kindly provided by Ralf Kühn (GSF National Research Centre for Environment and Health, Institute of Developmental Genetics, Neuherberg, Germany).

We subcloned a Cre-conditional FLEX (flip-excision) cassette (*Schnütgen et al., 2003*) into pRMCE, and inserted a strong CAG promoter (CMV immediate early enhancer/modified chicken β-actin promoter, from Addgene Plasmid #1378) in front of the FLEX-cassette to create pRMCE-CAG-Flex. The coding sequence of hM4Di-mKateT was inserted into the FLEX cassette in reverse orientation to the promoter (*Figure 1—figure supplement 2A*). Finally, a rabbit globulin polyA cassette

including stop codons in every reading frame was placed downstream of the FLEX cassette, in the same direction as hM4Di, in order to prevent unintended transcriptional read-through from potential endogenous promoters. The construct was completely sequenced before ES cell electroporation.

Electroporation of the RMCE construct together with a plasmid encoding C31int was performed by the transgene facility of the 'Research Institute for Experimental Medicine' (FEM, Charité, Berlin) according to published protocols (*Hitz et al., 2009*; *Hitz et al., 2007*). Recombinant clones were selected by incubation with 140 µg/ml G418 for at least 7 days. To activate hM4Di expression by recombination of the FLEX switch, selected clones were further transfected transiently with pCAG-Cre-EGFP using Roti-Fect (Carl Roth, Karlsruhe, Germany). G418-resistant clones were analyzed by PCR for successful integration and recombination of the construct (*Figure 1—figure supplement 2B*), using the following primers:

GT001 PGK3'-fw: CACGCTTCAAAAGCGCACGTCTG;
GT002 Neo5'-rev: GTTGTGCCCAGTCATAGCCGAATAG;
GT005 PolyA-fw: TTCCTCCTCTCCTGACTACTCC;
GT006 Rosa3'-rev: TAAGCCTGCCCAGAAGACTC;
GT013 hM4Di3'rec-rev: CAGATACTGCGACCTCCCTA

After verification of correct integration and functional FLEX-switch recombination, we generated chimeras by blastocyst injection of I3 ES cells. Heterozygous offsprings were mated with a Flpe deleter mouse line in order to remove the neomycin resistance cassette by Flp-mediated recombination.

Mice homozygous for the Rosa-CAG-FLEX-hM4Di-mKateT allele are viable and fertile and show now obvious phenotype. Importantly, application of CNO to these mice does not induce any behavioral effects. Homozygous Cre-conditional hM4Di transgenic mice and Sst-Cre mice (*Taniguchi et al., 2011*) were maintained on a C57Bl/6 genetic background and were bred to obtain heterozygous Sst-Cre / hM4Di offsprings.

## Drugs

Urethane (U2500, Merck), Fenfluramine ((+)-Fenfluramine hydrochloride, F112, Merck), 5-HT (Serotonin creatinine sulfate monohydrate, H7752, Merck), m-CPBG (1-(3-Chlorophenyl)biguanide hydrochloride, C144, Merck), tropisetron (Tropisetron hydrochloride,Y0000616, Sigma), WAY-100635 (W108, Merck), $\alpha$-Methylserotonin ($\alpha$-Methylserotonin maleate salt, M110, Merck), MDMA (($\pm$)3,4-methylenedioxymethamphetamine, 64057-70-1, Merck), CNO (Clozapine *N*-oxide dihydrochloride, 6329, Tocris) were dissolved in water for in vitro application and in 0.9% normal saline for in vivo application. Ketanserin (Ketanserin (+)-tartrate salt, S006, Merck) was dissolved in Dimethyl sulfoxide (DMSO).

## Surgery and in vivo recording

Before recordings mice were briefly anesthetized with isofluorane (2%) and then injected intraperitoneally with urethane (1.2 g/kg, Sigma Aldrich, Munich, Germany). The level of anesthesia was maintained so that hindlimb pinching produced no reflex movement and supplemental doses of urethane (0.2 g/kg) were delivered as needed. Upon cessation of reflexes the animals were mounted on a stereotaxic frame (Kopf Instruments, Tujunga, California), and body temperature was maintained at 38°C. The scalp was removed, and the skull was cleaned with saline solution. A craniotomy was performed at +3 mm ML, −3 mm AP, +3.25 mm DV to reach mEC.

Extracellular recordings from EC (*Figures 1–2*) were performed using a Cambridge Neurotech (Cambridge, United Kingdom) silicon probe (64-channels) (n = 15) or 32-channels (n = 3). The recording electrode was painted with the fluorescent dye DiI (Thermo Fisher Scientific, Schwerte, Germany) and then slowly lowered into the craniotomy using micromanipulators (Luigs and Neumann, Ratingen, Germany) at a 25° angle AP (toward the posterior side of the brain). The exposed brain was kept moist using saline solution. A ground wire connected to the amplifier was placed in the saline solution covering the skull to eliminate noise. Brain signals were recorded using a RHD2000 data acquisition system (Intan Technologies, Los Angeles, California) and sampled at 20 kHz. Recording quality was inspected on-line using the open-source RHD2000 Interface Software. A supplementary dose of urethane (0.2 g/kg) was injected right before the start of the recording to standardize anesthesia level across different experiments and avoid the arise of theta activity in the first 30 min of

recording. Recordings began after a 10 min waiting period in which clear upstates could consistently be seen at a regular frequency.

## In vivo analysis

We selected the channel to use for upstate detection based on the standard deviation (STD) of the trace during baseline (first 5 min of recording): the channel with the highest STD was selected, as larger voltage deflection increases detection algorithm accuracy. Given the highly synchronous nature of SOs (*Figure 3—figure supplement 1*) the spatial location of the channel selected was not considered. Upstates were detected comparing threshold crossing points in two signals: the delta-band filtered signal (0.5–4 Hz) and the population spike activity. Candidate upstates were identified in the delta-band filtered signal using two dynamic thresholds 'a' and 'b':

$$a = m + \frac{\sigma}{1.5}$$

$$b = m + \frac{\sigma}{0.8}$$

where σ is the standard deviation of the signal during the first five minutes of recording (baseline) and *m* is the centered moving median calculated using 60 s windows (Matlab function *movmedian*). The median was used instead of the mean to account for non-stationaries in the data. A candidate upstate was identified at first using the threshold crossings of the signal compared to 'a': candidates shorter than 200 ms were deleted and multiple candidates occurring within a window of 300 ms were joined together. Subsequently, the threshold 'b' was used to separate upstates occurring in close proximity: if the signal within one candidate crossed the threshold 'b' in more than one point then the candidate upstate was split in two at the midpoint between the two threshold crossings. Candidate upstates were finally confirmed if the population spike activity (calculated in 100 ms windows) within the candidate crossed a threshold of 1 σ (calculated during the baseline).

## Units detection and classification

Spike detection was performed offline using the template-based algorithm Kilosort2 https://github.com/MouseLand/Kilosort2; *Filippo, 2021a*; copy archived at swh:1:rev:a1fccd9abf13ce5dc3340-fae8050f9b1d0f8ab7a, with the following parameters:

- ops.fshigh = 300
- ops.fsslow = 8000
- ops.minfr_goodchannels = 0
- ops.Th = [8 4]
- ops.lam = 10
- ops.AUCsplit = 0.9
- ops.minFR = 1/1000
- ops.momentum = [20 400]
- ops.sigmaMask = 30
- ops.ThPre = 8
- ops.spkTh = −6
- ops.nfilt_factor = 8
- ops.loc_range = [3 1]
- ops.criterionNoiseChannel = 0.2
- ops.whiteningrange = 32
- ops.ntbuff = 64

Manual curation of the results was performed using Phy https://github.com/cortex-lab/phy; *Filippo, 2021b*; copy archived at swh:1:rev:6ffe05a559bd0302e98ec60d5958ace719544713. Each Isolated unit satisfied the following two criteria: Refractory period (2 ms) violations < 5%, fraction of spikes below detection threshold (as estimated by a gaussian fit to the distribution of the spike amplitudes)<15%. Units with negative maximal waveform amplitude were further classified as putative excitatory if the latency (TP latency) was >0.55 ms or putative inhibitory when TP latency <0.55 ms. The value 0.55 was chosen in accordance with previous studies (*Senzai et al., 2019*; *Antoine et al., 2019*). In pharmacological classification, units were classified as 'activated' if

their firing rate in the 25 min following drug injection was 2 σ (standard deviation) above the baseline rate for at least 5 min. Remaining units were pulled together in the category 'non-activated'.

## Cross-correlogram analysis

Cross-correlogram based connectivity analysis was performed for every unit to identify inhibitory connections. Units with a spiking rate smaller than 0.3 spikes/s were discarded from the analysis. We used total spiking probability edges (TPSE) algorithm https://github.com/biomemsLAB/TSPE; *Filippo, 2021c*; copy archived at swh:1:rev:b780c753039a2f48201a6bb77dd8f5e65551a845, *De Blasi et al., 2019* to identify in a computationally efficient manner putative inhibitory connections between units and all clusters recorded. The parameters used were:

- d = 0,
- neg_wins = [2, 3, 4, 5, 6, 7, 8],
- co_wins = 0,
- pos_wins = [2, 3, 4, 5, 6],
- FLAG_NORM = 1.

The connectivity vectors of each unit resulting from TSPE were sorted by inhibition strength. The top 20 connections were further analyzed using custom Matlab (RRID:SCR_001622) code. A connection was classified as inhibitory if the cross correlogram values (x) were smaller than the mean of x by more than one standard deviation (x < mean(x) – std(x)) in at least four consecutive bins (bin size = 1 ms) in a window 4–9 ms after the center of the cross-correlogram.

## Slice preparation

We prepared acute near horizontal slices (~15° off the horizontal plane) of the mEC from C57Bl/6 mice. Animals were decapitated following isoflurane anesthesia. The brains were quickly removed and placed in ice-cold (~4° C) ACSF (pH 7.4) containing (in mM) 85 NaCl, 25 NaHCO$_3$, 75 Sucrose, 10 Glucose, 2.5 KCl, 1.25 NaH$_2$PO$_4$, 3.5MgSO$_4$, 0.5 CaCl$_2$, and aerated with 95% O$_2$, 5% CO$_2$. Tissue blocks containing the brain region of interest were mounted on a vibratome (Leica VT 1200, Leica Microsystems), cut at 400 μm thickness, and incubated at 35°C for 30 min. The slices were then transferred to ACSF containing (in mM) 85 NaCl, 25 NaHCO$_3$, 75 Sucrose, 10 Glucose, 2.5 KCl, 1.25 NaH$_2$PO$_4$, 3.5 MgSO$_4$, 0.5 CaCl$_2$. The slices were stored at room temperature in a submerged chamber for 1–5 hr before being transferred to the recording chamber.

## In vitro recording

In order to perform whole-cell recordings slices were transferred to a submersion style recording chamber located on the stage of an upright, fixed-stage microscope (BX51WI, Olympus) equipped with a water immersion objective (×60, Olympus) and a near-infrared charge-coupled device (CCD) camera. The slices were perfused with ACSF (~35°C bubbled with 95% O$_2$–5% CO$_2$) at 3–5 ml/ min to maintain neuronal health throughout the slice. The ACSF had the same composition as the incubation solution except for the concentrations of calcium and magnesium, which were reduced to 1.2 and 1.0 mM, respectively. Recording electrodes with impedance of 3–5 MΩ were pulled from borosilicate glass capillaries (Harvard Apparatus, Kent, UK; 1.5 mm OD) using a micropipette electrode puller (DMZ Universal Puller). The intracellular solution contained the following (in mM): 135 K-gluconate, 6 KCl, 2 MgCl$_2$, 0.2 EGTA, 5 Na2- phosphocreatine, 2 Na2-ATP, 0.5 Na2-GTP, 10 HEPES buffer, and 0.2% biocytin. The pH was adjusted to 7.2 with KOH. Recordings were performed using Multiclamp 700A/B amplifiers (Molecular Devices, San Jose, California). The seal resistance was >1 GΩ. Capacitance compensation was maximal and bridge balance adjusted. Access resistance was constantly monitored. Signals were filtered at 6 kHz, sampled at 20 kHz, and digitized using the Digidata 1550 and pClamp 10 (Molecular Devices, San Jose, California). Activation light was delivered by a 460 nm laser (DPSS lasers, Santa Clara, California) using a 460–480 nm bandpass excitation filter. Stimulation consisted of 500 ms pulses at 1 Hz.

Stimulation experiments were performed using a bipolar micro-electrode (glass pipette filled with ACSF solution, wrapped by a fine grounding wire) connected to an isolated voltage stimulator (ISO-Flex, A.M.P.I., Israel). A 4x objective (Olympus) was used to visually guide the stimulating electrode into the mEC. Stimulation power was adjusted to achieve consistent upstate generation during baseline (>95%). Each stimulus had a duration of 50 μs and the inter-stimulus interval was 8–10 s.

## In vitro analysis

In vitro upstates were detected in Matlab using an algorithm similar to the one described in the in vivo analysis method section. We used a coincident detection in two signals. In multicellular recordings, we used the membrane potential of two cells, and in single-cell recordings, we used membrane potential and the envelope of the gamma filtered trace (50–250 Hz), as upstates are characterized by an increase in gamma activity (*Neske, 2015*).

The baseline condition was defined as the last 120 s before drug application, while the post-drug application condition was defined as the 120 s of recording after drug application (Total recording duration: 600 s).

Excitatory (Exc), fast-spiking (FS), and low-threshold spiking (LTS) neurons were classified using Gaussian mixture models (GMM) with a soft clustering approach in Matlab. Input resistance ($R_{in}$), $\Delta$after-hyperpolarization ($\Delta$AHP), sag, rheobase, spike width and resting potential (RP) were extracted from each neuron and used in the classification. The first two components of the principal component analysis (PCA) were used to fit the data to a Gaussian mixture model distribution. Initial values were set according to the k-means algorithm with centroid positioned at x and y position: 5, 0; −15,–15; −15, 10. This centroid were placed according to the loadings of the PCA to identify three clusters with the following main features:

- Cluster 1 (putative Exc): high spike width, low AHP, low rheobase.
- Cluster 2 (putative FS): low spike width, low SAG, high rheobase, low $R_{in}$.
- Cluster 3 (putative LTS): low spike width, high SAG, high AHP, high $R_{in}$.

Covariance matrices were diagonal and not shared. Neurons with a posterior probability of belonging to any of the three clusters of <90% were discarded from further analysis (1/49).

While the majority of Sst-interneurons display LTS features, a minority (~10%) belong to the FS group (*Urban-Ciecko et al., 2015*). To distinguish FS and LTS interneurons in the Sst-Td Tomato mice, we employed the GMM with posterior probability threshold of 90%. Only LTS Sst neurons were considered for further analysis.

## Histological analysis

For the postmortem electrode track reconstructions of the in vivo recordings, mice were not perfused; rather, brains were extracted from the skull, post-fixed in 4% PFA overnight at 4°C and afterwards cut with a vibratome (Leica Microsystems, Wetzlar Germany) in 100 μM thick sequential sagittal slices. Images were taken using a 1.25x objective and stitched together using the microscope software (BX61, Olympus). Afterwards, we used AllenCCF code https://github.com/cortex-lab/allenCCF; *Filippo, 2021d*; copy archived at swh:1:rev:80fb1326ccaf6ae765944173c7465f650ade-babe, to identify electrode shank location (*Shamash et al., 2018*).

For the anatomical reconstructions of recorded cells in vitro, brain slices were fixed with 4% paraformaldehyde in 0.1 M phosphate buffer (PB) for at least 24 hr at 4°C. After being washed three times in 0.1 M PBS, slices were then incubated in PBS containing 1% Triton X-100% and 5% normal goat serum for 4 hr at room temperature (RT). To visualize biocytin-filled cells we used Streptavidin Alexa 488 conjugate (1:500, Invitrogen Corporation, Carlsbad, CA, RRID:AB_2315383). WFS1 (1:1000, Rabbit, Proteintech, IL, USA, RRID:AB_2880717) was used in a subset of experiments to visualize the L2/L3 border, and Sst (1:1000, Rat, Bachem, Switzerland, RRID:AB_2890072) was used in the 5-HT$_{2A}$R localization analysis. Slices were incubated with primary antibodies for 48 hr at RT. After rinsing two times in PBS, sections were incubated in the PBS solution containing 0.5% Triton X-100, Alexa fluor 488, Alexa fluor 555 and Alexa fluor 647 (Invitrogen Corporation, Carlsbad, CA) according to the number of antibodies used. Slices were mounted in Fluoroshield (Sigma-Aldrich) under coverslips 2–3 hr after incubation with the secondary antibodies and stored at 4°C.

Labeled cells were visualized using 20x or 40x objectives on a confocal microscope system (SP8, Leica, RRID:SCR_018169). For the 5-HT$_{2A}$R localization analysis, images of the whole EC were acquired and stitched together using the auto stitching method, without smoothing. Z stacks were acquired every 30 μM. The image stacks obtained were registered and combined in Fiji (RRID:SCR_002285) to form a montage of the sections. Cell counting was executed using the Fiji multi-point tool. X-Y-Z coordinates of each 5-HT$_{2A}$R-EGFP-positive cell were exported to Matlab and subsequently, using custom written code in Matlab, we semi-automatically inspected each cell for colocalization between EGFP(5-HT$_{2A}$R) and Sst.

## Statistical analysis

All datasets were tested to determine normality of the distribution either using D'Agostino-Pearson omnibus normality test or Shapiro-Wilk normality test. Student's t-test and one-way ANOVA were used for testing mean differences in normally distributed data. Wilcoxon matched-pairs signed rank test and Kruskal-Wallis were used for non-normally distributed datasets. Dunn-Sidak multiple comparison test was used to compare datasets with three or more groups. Kolmogorov-Smirnov test was used to compare cumulative distributions. Statistical analysis was performed using Prism (6.01, RRID:SCR_002798) and Matlab. All data are expressed as mean ± SEM. Asterisks in figures represent p-values smaller than 0.05, unless stated otherwise in the legend.

## Additional information

### Funding

| Funder | Grant reference number | Author |
|---|---|---|
| Bundesministerium für Bildung und Forschung | SFB1315-327654276 | Dietmar Schmitz |
| Deutsche Forschungsgemeinschaft | SPP1926 | Benjamin R Rost |
| Deutsche Forschungsgemeinschaft | SPP1665 | Dietmar Schmitz |
| Deutsche Forschungsgemeinschaft | 01GQ1420B | Dietmar Schmitz |
| Deutsche Forschungsgemeinschaft | HA5741/5-1 | Christoph Harms |
| Deutsche Forschungsgemeinschaft | TRR295 | Christoph Harms |
| Bundesministerium für Bildung und Forschung | 01EO0801 | Christoph Harms |
| NeuroCure Exzellenzcluster | EXC-2049 - 390688087 | Dietmar Schmitz |
| Charité – Universitätsmedizin Berlin | BIH Fellowship | Prateep Beed |

The funders had no role in study design, data collection and interpretation, or the decision to submit the work for publication.

### Author contributions

Roberto de Filippo, Conceptualization, Data curation, Software, Formal analysis, Investigation, Visualization, Writing - original draft, Project administration, Writing - review and editing; Benjamin R Rost, Conceptualization, Resources, Methodology, Writing - review and editing; Alexander Stumpf, Claire Cooper, Investigation, Writing - review and editing; John J Tukker, Resources, Methodology, Writing - review and editing; Christoph Harms, Resources, Writing - review and editing; Prateep Beed, Conceptualization, Supervision, Investigation, Methodology, Writing - review and editing; Dietmar Schmitz, Conceptualization, Resources, Supervision, Funding acquisition, Writing - original draft, Writing - review and editing

### Author ORCIDs

Roberto de Filippo https://orcid.org/0000-0002-4085-9114
John J Tukker http://orcid.org/0000-0002-4394-199X
Christoph Harms http://orcid.org/0000-0002-2063-2860
Dietmar Schmitz https://orcid.org/0000-0003-2741-5241

### Decision letter and Author response

Decision letter https://doi.org/10.7554/eLife.66960.sa1
Author response https://doi.org/10.7554/eLife.66960.sa2

## Additional files

### Supplementary files
• Transparent reporting form

### Data availability

A database for the *in vivo* data is available at http://dx.doi.org/10.6084/m9.figshare.14229545. Custom code used for the analysis is available at https://github.com/Schmitz-lab/De-Filippo-et-al-2021 (copy archived at https://archive.softwareheritage.org/swh:1:rev:eda027f20c4f267762bc46c47f976ff-f49a0aa84/). Code used for spike sorting is available at https://github.com/MouseLand/Kilosort2 (copy archived at https://archive.softwareheritage.org/swh:1:rev:a1fccd9abf13ce5dc3340-fae8050f9b1d0f8ab7a/). Code used for manual curation of sorted spikes is available at https://github.com/cortex-lab/phy (copy archived at https://archive.softwareheritage.org/swh:1:rev:6ffe05a559bd0302e98ec60d5958ace719544713/). Code used to estimate neuronal connectivity is available at https://github.com/biomemsLAB/TSPE (copy archived at https://archive.softwareherit-age.org/swh:1:rev:b780c753039a2f48201a6bb77dd8f5e65551a845/).

The following dataset was generated:

| Author(s) | Year | Dataset title | Dataset URL | Database and Identifier |
|---|---|---|---|---|
| de Filippo R | 2021 | Somatostatin interneurons activated by 5-HT2A receptor suppress slow oscillations in medial entorhinal cortex | http://dx.doi.org/10.6084/m9.figshare.14229545 | figshare, 10.6084/m9.figshare.14229545 |

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
