## [Decision Letter]

**Acceptance summary:**

This paper identifies a mechanism through which serotonin, one of the brain's major neuromodulators, affects network activity, through 5HT2a-mediated activation of a major subclass of GABAergic interneurons – Somatostatin-positive interneurons. These findings advance our understanding of how brain state affects cortical circuits through the action of specific cell classes.

**Decision letter after peer review:**

[Editors’ note: the authors submitted for reconsideration following the decision after peer review. What follows is the decision letter after the first round of review.]

Thank you for submitting your work entitled "Serotonin suppresses slow oscillations by activating somatostatin interneurons via the 5-HT_2A_ receptor" for consideration by *eLife*. Your article has been reviewed by four peer reviewers, one of whom is a member of our Board of Reviewing Editors, and the evaluation has been overseen by a Senior Editor. The following individual involved in review of your submission has agreed to reveal their identity: Antonio Fernandez-Ruiz (Reviewer #3).

Our decision has been reached after consultation between the reviewers. Based on these discussions and the individual reviews below, we regret to inform you that your work will not be considered further for publication in *eLife* at present. However given the interest of the reviewers in the topic, we welcome a resubmission to e*Life* if it is possible to run additional experiments requested by the reviewers; this submission would be treated as a new submission but would be send to the same reviewers. Although the reviewers made several positive comments, they identified several major issues, in particular: 1) The interpretation and framing of the paper in terms of slow oscillations, which was questioned by several reviewers. 2) The selectivity of pharmacological manipulation (see Reviewer 2), and 3) Data analysis. The reviewers agreed that while recordings without anesthesia would enhance the manuscript, these experiments are beyond the scope of the paper, although this limitation needs to be discussed more clearly.

Reviewer #1:

This paper shows that slow oscillations in entorhinal cortex are affected by serotonin, in particular acting through 5HT2AR receptor, and that this effect is mediated by Somatostatin interneurons. Although the generality of this finding for other cortical areas and natural (i.e. not anesthesized) conditions remains to be established, I find this is a set of novel findings that could be important to understand control of brain state and also clinical applications. Critique:

1) A clear short-coming of the paper is the lack of recordings during naturally occurring slow waves. If it is reasonable to conduct these experiments, this would greatly enhance the paper. In particular potential effects of anesthesia on SSt interneurons need to be taken into account (see Adesnik and Scanziani, 2012).

2) A second critical question is whether there is at all an effect on slow oscillations. It appears to me that there is a massive suppression of spiking activity such that the circuit is essentially in a sustained downstate that is not at all representative of the desynchronized state as one normally finds it, and what would have happened with stimulating the Raphe Nucleus. So the conclusion that there's an effect on slow oscillations is not entirely clear to me. This pertains to Figure 1 and 4, for instance.

Would blocking 5HT2aRs during normal wake-to-sleep or arousal-to-SWS transitions abolish slow oscillations? If one would stimulate the Raphe and blocks 5HT2aR what would one predict? I understand these are difficult experiments to do, but does the paper bear relevance for these more naturalistic scenarios?

3) For completeness it would be useful to comment on the other GABAergic cells that have 5HT3Rs and explain why their contribution is not relevant for different pharmacological agents or more in general.

Reviewer #2:

The manuscript by De Filippo et al. presents a combined in vivo / in vitro study of the cellular mechanisms of network dynamics in the entorhinal cortex (EC). The main findings support 5-HT suppression of slow oscillatory activity, mediated by 5-HT2AR activation of SOM interneurons. This is a novel finding, potentially of general interest. My major concerns relate to the strength of the in vivo evidence and the framing of the study.

Concerns related to in vivo data

1) The authors present clear demonstration that the pharmacological agents used suppress the slow oscillation (SO) in EC. Given that none of these drugs are truly selective and that they are presented i.p., the authors provide no direct evidence that these effects are 1) mediated by 5-HT release and 2) mediated by 5-HT release in the entorhinal cortex. This can be addressed by preventing drug-mediated SO reduction with local application of 5-HT2AR antagonists directly in the EC.

2) The in vivo single unit data does not support the activation of SOM interneurons.

a) The activated neurons appear to include both putative inhibitory (fast spiking) and putative excitatory (non-fast spiking) neurons (Figure 2C), rather than an intermediate, potential SOM population. If the activated neurons include both FS inhibitory and non-FS excitatory neurons, one would expect TP latency of the population to be intermediate as shown in Figure 2B,C.

b) Given the inability to identify SOM interneurons based on waveform alone, additional characterization of the activated units is necessary. First, do the CCGs of the activated units indicate inhibitory connections? These data are notably missing (or not easily extracted from Figure 2—figure supplement 1).

c) Second, definitely identifying SOM interneurons in vivo would require some type of tagging.

Concerns related to the Introduction

3) Motivating this study by discussing psychological processes, psychiatric disorders, sleep/wake cycles or psychoactive drugs is highly tenuous and misleading. This study does not address any of these processes. Alternatively, the Introduction lacks even a superficial description of topics that are highly relevant to this study, including neuroanatomy and neural circuitry of the EC, physiological properties of GABAergic interneurons in the EC, previous studies of the slow oscillation and involvement of different neuronal cell types in the EC, and neuromodulation of different interneuron subtypes, especially known patterns of neuromodulatory receptor expression on SOM interneurons.

Reviewer #3:

The manuscript by De Filipo et al. presents a well-executed and novel study on the role of somatostatin positive (Som) interneurons in the generation of cortical slow oscillations (SO) in the entorhinal area. The authors employ a combination of mouse in vivo recordings under anesthesia, slice physiology, together with pharmacological and chemo/optogenetic manipulations to shown that entorhinal SO are suppressed by serotonin activation of Som interneurons. The present work provides new data on an interesting topic that can have broader implications to understand the relationship between neuromodulators, state-dependent brain oscillations and different interneuron subtypes. The experimental approaches are rigorous and adequately support the authors claims. However, the study has some limitations that cast doubts on the specificity of the effects reported. Additional analysis may be sufficient to clarify these doubts.

1) The authors need to show more convincing evidence that the effects 5-HT and the manipulations reported in Figure 1 are specific of the SO and not a general dampened of entorhinal activity. More extensive analysis of the current data will suffice for that purpose. Figure 1H suggests that there may be a reduction in power in other frequencies than δ, but is not clearly appreciated (using a log frequency axis will facilitate it). A power spectrum comparison before, during and after drug application is necessary to evaluate the changes in all frequencies. Entorhinal cortex displays prominent γ oscillations, even during sleep and urethane anesthesia, are those also affected? This should be specifically quantified. Under urethane there is an alternation between a SO and a slow theta (~3-4Hz) state. Are theta oscillations also affected by the pharmacological manipulations?

2) More comprehensive analysis of the effect of the different manipulations on SO reported through the manuscript should be performed, not only the rate of up states. For example, distribution of durations, inter-event intervals, amplitude of the remaining up states, etc. These will help to better understand the nature of the effects of serotonin release on SO.

3) Expanding the analysis and discussion of the contribution of PV interneurons will help establish the specificity of the role of Som interneurons. The results of the experiment with hM4Di-PV are encouraging in this regard. The authors do have fast-spiking, narrow-waveform cells (thus putative PV+ interneurons) in their dataset; however, there is very little discussion about this through the text. The results in Figure 4 provide evidence of the ability of Som interneurons to suppress SO. However, I would expect a very similar effect from PV activation. Unless the authors have evidence of the opposite, they should acknowledge this.

Reviewer #4:

In this manuscript, De Filippo et al. identify somatostatin-containing interneurons (SOM) of the entorhinal cortex (EC) as targets of serotonergic neurons in the dorsal raphe and describe their involvement in the mediation of slow oscillations in the EC, using an array of techniques such as pharmacology, optogenetics and DREADD.

The manuscript is well and the analysis is solid. While there is no doubt this study demonstrates convincingly the involvement of SOM neurons in the regulation of SO, it fails to convincingly rule out other cell types which could equally contribute to the cortical dynamics under investigation. In that respect, I am left with some lingering doubts as to whether or not something was missed in the study and would therefore appreciate some clarifications on the part of the authors (see major point below).

1) Some previous studies (for example, Ferezou et al., 2002) have shown that VIP/CCK GABAergic cells express 5-HT3 receptors, which are ionotropic receptors and can therefore respond very fast (with a depolarization, therefore an activation) to serotonergic neuromodulation. How come the authors do not mention this study (and others) and could they give an explanation as to why none of these neocortical cells (presumably also present in the EC?) were identified in their study as activated? Alternatively, using mCPBG in a similar fashion to how α-methyl5-HT was used in Figure 3 to rule out involvement of 5-HT3R would be valuable. The authors also mention the fact that only 11% of the cells expressing 5-HT2AR promoter were SOM. Could they please elaborate on the remaining 89% of cells?

2) Studies in the somatosensory cortex of awake mice (presumably therefore at a time when 5-HT release is highest) showed that L2/3 SOM neurons were not correlated with other neuronal cell types at the level of their membrane potentials and that they hyperpolarized and therefore reduce their spiking during active wake (when the mouse movedits whiskers (Gentet et al., 2012, Muñoz et al., 2017)). The authors should discuss their findings in the light of these papers considering that they argue that serotonin activates SOM neurons in the EC and 5-HT release is highest during wake, and therefore presumably, even more so during active wake.

---

## [Author Response]

[Editors’ note: the authors resubmitted a revised version of the paper for consideration. What follows is the authors’ response to the first round of review.]

Reviewer #1:This paper shows that slow oscillations in entorhinal cortex are affected by serotonin, in particular acting through 5HT2AR receptor, and that this effect is mediated by Somatostatin interneurons. Although the generality of this finding for other cortical areas and natural (i.e. not anesthesized) conditions remains to be established, I find this is a set of novel findings that could be important to understand control of brain state and also clinical applications. Critique:1) A clear short-coming of the paper is the lack of recordings during naturally occurring slow waves. If it is reasonable to conduct these experiments, this would greatly enhance the paper. In particular potential effects of anesthesia on SSt interneurons need to be taken into account (see Adesnik and Scanziani, 2012).

Recording during natural sleep together with application of either MDMA or fenfluramine (Fen) would be problematic as both these compounds act as sleep suppressant in both rodents and humans (Myers et al., 1993, Fornal and Radulovacki, 1983, Balogh et al., 2004, Randall et al., 2009). We agree that more effort should be put in studying the effect of 5-HT on Som interneurons in vivo and we plan on doing so in a following work in awake head-fixed condition. In accordance with the reviewing and senior editors’ opinions, we believe that these experiments would go beyond the aims of our current study. We modified the text to explicitly state that the absence of data in naturalistic conditions represent a limitation of our work.

2) A second critical question is whether there is at all an effect on slow oscillations. It appears to me that there is a massive suppression of spiking activity such that the circuit is essentially in a sustained downstate that is not at all representative of the desynchronized state as one normally finds it, and what would have happened with stimulating the Raphe Nucleus. So the conclusion that there's an effect on slow oscillations is not entirely clear to me. This pertains to Figure 1 and 4, for instance.Would blocking 5HT2aRs during normal wake-to-sleep or arousal-to-SWS transitions abolish slow oscillations? If one would stimulate the Raphe and blocks 5HT2aR what would one predict? I understand these are difficult experiments to do, but does the paper bear relevance for these more naturalistic scenarios?

The choice of the word “desynchronization” was unfortunate as what we observe in our data does not reflect in vivo awake classic “desynchronization”(Chen et al., 2015), we revised the Discussion to claim instead that 5-HT suppresses synchronized activity. We agree that the effect we see, both in vivo and in vitro, can be described as a general dampening of spiking activity. We consider suppression of upstates to be equal to suppression of slow oscillations (SO), as we do not believe that a constant downstate can be categorized as slow oscillation. Our usage of the terms upstate and downstate follow what, in our view, is the field convention (Harris and Thiele, 2011), some works use a different definition (see (Niethard et al., 2018)).

We use SO as a proxy to understand the effect of 5-HT on networks of cortical neurons but we do not aim to make conclusive statements about the relevance of our findings in natural sleep. For example, during natural sleep (both slow-wave sleep and REM) the level of 5-HT in the brain are at the lowest (Oikonomou et al., 2019, Unger et al., 2020, McGinty and Harper, 1976), thus, the increase of 5-HT during sleep would be of itself an artificial condition. We investigated in a new set of experiments (Author response image 1 and Figure 3) the effect of Raphe 5-HT neurons optogenetic activation during anesthesia in the same cortical region (medial entorhinal cortex). We did not see suppression of SO but only a decrease in spiking rate during upstates. Decrease in spiking following the same manipulation has been reported in a previous study in somatosensory and motor areas (Grandjean et al., 2019). Our explanation for this difference is that the amount of 5-HT released using pharmacological or optogenetic release is probably considerably different. As we explain in the revised text, MDMA and Fen (5 mg/kg) have been reported to induce a ~20-fold increase in peak 5-HT concentration both in monkey and rodent brain (Udo de Haes et al., 2006, Gołembiowska et al., 2016). On the other hand, light activated 5-HT neurons in ePet mice seems to fire at a physiological rate (≈ 3 Hz) (Ranade and Mainen, 2009, Sakai and Crochet, 2001) even when stimulated at higher frequencies. (Grandjean et al., 2019). In consequence it is likely that optogenetic activation is not eliciting the release of enough 5-HT to cause SO suppression and Som interneurons activation during anesthesia. We decided to include these data as we think that the different effect of pharmacological and optogenetic release of 5-HT might be of interest to the community especially considering the potential usage of MDMA as medication for post-traumatic stress disorder (Mithoefer et al., 2019, Inserra et al., 2021).

We do not believe that 5-HT is solely responsible for SO suppression between sleep to wake transition, it is known that acetylcholine (Chen et al., 2015) and noradrenaline (Steriade et al., 1993) can both suppress SO, and both of them are also synced with the wake cycle (higher levels during wake compared to slow wave sleep (Watson et al., 2012, Mitchell and Weinshenker, 2010). This makes pharmacology experiments in sleeping condition difficult to interpret in our opinion. In the future we will focus on 5-HT2A pharmacological manipulations in vivo awake conditions, we believe that these experiments go beyond the aims of the current study.

**Author response image 1. sa2fig1:** SO are not suppressed by optogenetic activation of Raphe 5-HT neurons during anesthesia. (A) DRN immunohistochemistry image showing ChR2-YFP infection in DRN serotonergic neurons. 5-HT in red, YFP in green. Scale bars: 50 µm. (B) 3D visualization of microelectrodes location for all experiments and optic fiber (black) location. EC represented in grey, dorsal Raphe nuclei (DRN) in pink. (C) Top: upstates raster plot. Bottom: upstates density using Gaussian kernel density estimation. (D) Histogram of upstates incidence during baseline and laser stimulation (n animals = 4onedr, baseline: 2.25 ± 0.13 upstates/10s, laser: 2.11 ± 0.14 upstates/10s). (E) Left: normalized spike population activity during baseline (black) and laser stimulation (blue) in mEC L3/5 (n = 108, p < 0.05, unpaired t-test with Holm-Šidák correction). Right: heatmap showing average spike rate difference per cluster between stimulation and baseline during upstates, some units show reduced spiking during stimulation. (F) Left: normalized spike population activity during baseline (black) and laser stimulation (blue) in CA1 (n = 140, p > 0.05, unpaired t-test with Holm-Šidák correction). Right: heatmap showing average spike rate difference per cluster between stimulation and baseline during upstates, no clear difference in spiking activity.

3) For completeness it would be useful to comment on the other GABAergic cells that have 5HT3Rs and explain why their contribution is not relevant for different pharmacological agents or more in general.

We performed a new set of in vitro experiment to complement some already available preliminary data on 5-HT_3_R and we can confirm that in vitro application of either 5-HT_3_R agonist and antagonist do not have any effect on SO (Figure 3—figure supplement 4). We integrated these results in the main text.

Reviewer #2:The manuscript by De Filippo et al. presents a combined in vivo / in vitro study of the cellular mechanisms of network dynamics in the entorhinal cortex (EC). The main findings support 5-HT suppression of slow oscillatory activity, mediated by 5-HT2AR activation of SOM interneurons. This is a novel finding, potentially of general interest. My major concerns relate to the strength of the in vivo evidence and the framing of the study.Concerns related to in vivo data1) The authors present clear demonstration that the pharmacological agents used suppress the slow oscillation (SO) in EC. Given that none of these drugs are truly selective and that they are presented i.p., the authors provide no direct evidence that these effects are 1) mediated by 5-HT release and 2) mediated by 5-HT release in the entorhinal cortex. This can be addressed by preventing drug-mediated SO reduction with local application of 5-HT2AR antagonists directly in the EC.

To address whether the suppression we see in vivo is selectively mediated by 5-HT we performed new in vivo anesthetized recordings in ePet-cre mice where we activated, using an optogenetic approach, 5-HT neurons in the dorsal raphe nucleus (DRN) (Author response image 1/Figure 3), this experimental paradigm enabled us to have better temporal control on 5-HT release.

Our in vitro data show that suppression of SO is not mediated exclusively by 5-HT_2A_R and this is reflected by the fact that even in presence of ketanserin (5-HT_2A_R antagonist) 5-HT is still able to significantly suppress SO (Figure 4—figure supplement 3 B-E, ≈ 60% incidence reduction). This, however, is not surprising because in mEC pyramidal cells are known to express 5-HT_1A_R, a receptor that causes hyperpolarization via activation of G protein-coupled inwardly rectifying potassium (GIRK) channels (Schmitz et al., 1998, Chalmers and Watson, 1991). We decided not to inject ketanserin together with MDMA/Fen in vivo because the potential action of 5-HT_1A_R would make the result difficult to interpret. For example, the amount of 5-HT released by these compounds (MDMA/Fen), vastly greater than physiological, could activate 5-HT_1A_R to an extent that would suppress SO without the need of 5-HT_2A_R.

In contrast to the application of MDMA/Fen, after optogenetic 5-HT release, we did not observe suppression of upstate incidence but only a reduction in spiking rate during upstates. A possible explanation might be that the amount of 5-HT released via optogenetic activation is considerably less compared to the pharmacological release. These results prevent us to claim that the effect we see in vivo is due solely to 5-HT release, nonetheless, we like to underline that Fen has been used before as a 5-HT selective releaser with the specific purpose of disentangle the effect of MDMA (Heifets et al., 2019). To strengthen the link between suppression and 5-HT2AR we performed a new set of in vitro experiments showing that ketanserin (5-HT_2A_R antagonist) dampens the suppressive power of fenfluramine (Figure 3—figure supplement 1). In this revised version we are careful to avoid strong statement such as “5-HT is sufficient to suppress slow oscillations in vivo”, nonetheless, considering our in vitro data, we believe it is legitimate to state that 5-HT plays a role in the suppression. We discuss this in the penultimate paragraph of the Discussion. We agree that more effort should be put in studying the effect of 5-HT on Som interneurons in vivo and we plan on doing this in a following work in awake head-fixed condition. We would prefer to focus on more naturalistic conditions to disentangle the relationship between Som interneurons and 5-HT especially because anesthesia is known to be a confounding factory particularly in the case of Som interneurons (Adesnik et al., 2012).

We believe, however, that these experiments would go beyond the aims of our current study as they would also require a new animal license.

2) The in vivo single unit data does not support the activation of SOM interneurons.a) The activated neurons appear to include both putative inhibitory (fast spiking) and putative excitatory (non-fast spiking) neurons (Figure 2C), rather than an intermediate, potential SOM population. If the activated neurons include both FS inhibitory and non-FS excitatory neurons, one would expect TP latency of the population to be intermediate as shown in Figure 2B,C.b) Given the inability to identify SOM interneurons based on waveform alone, additional characterization of the activated units is necessary. First, do the CCGs of the activated units indicate inhibitory connections? These data are notably missing (or not easily extracted from Figure 2—figure supplement 1).c) Second, definitely identifying SOM interneurons in vivo would require some type of tagging.

a) We agree that our data do not prove conclusively the involvement of Som Interneurons in SO suppression in vivo. We also agree that if the activated neurons were to be equally putative FS and putative excitatory the final TP latency average would be in between these two groups. To understand whether this is the case we looked at the cumulative probability distribution (Figure 2B, middle panel) where we can see a clear difference between the three groups. This is reflected by the fact that units with TP latency between 0.4 and 0.7 ms have much higher likelihood (from ~40% to ~80%, red line, right y-axis) of being part of the “activated” group compared to units with any other TP latency (Figure 3B bottom). If the “activated” units were to be equally distributed across TP latencies we should not have such peak in our opinion. We added a clarification in the text. In conclusion, we believe that these data support our in vitro data regarding Som interneurons.

b) CCG based connectivity analysis requires a minimum baseline firing rate to be effective, now we explain this detail in the legend and not only in the Materials and methods section. The threshold we used is 0.3 spikes/s, units missing from the analysis have a lower firing rate. We acknowledge that this analysis does not help in characterizing the intermediate population as inhibitory therefore we are open to remove the image as superfluous.

c) We agree and we revised the text to clarify that our data do not demonstrate conclusively the involvement of Som interneurons in vivo. We consider our in vitro experiment to show conclusively the involvement of Som interneurons activated by 5-HT_2A_ in SO suppression. Our in vivo data merely support this notion as Som interneurons in vivo might have a TP latency in between FS and excitatory neurons (Trainito et al., 2019). Optotagging in vivo would answer the question conclusively however the location of the entorhinal cortex (EC) makes this experiment particularly challenging, to have a decent chance of finding responsive units we should use Neuropixels probes together with an external optic fiber. The number of responsive units In Sst-cre mice found by the Allen institute using Neuropixels has also discouraged us from performing optotagging (8/350 units, example found at https://allensdk.readthedocs.io/en/latest/_static/examples/nb/ecephys_optotagging.html). Moreover anesthesia is a known confounding factor especially in the case of Som interneurons (Adesnik et al., 2012), for this reason we would prefer to focus on non-anesthetized conditions, something we plan to do in our next work.

Concerns related to the Introduction3) Motivating this study by discussing psychological processes, psychiatric disorders, sleep/wake cycles or psychoactive drugs is highly tenuous and misleading. This study does not address any of these processes. Alternatively, the Introduction lacks even a superficial description of topics that are highly relevant to this study, including neuroanatomy and neural circuitry of the EC, physiological properties of GABAergic interneurons in the EC, previous studies of the slow oscillation and involvement of different neuronal cell types in the EC, and neuromodulation of different interneuron subtypes, especially known patterns of neuromodulatory receptor expression on SOM interneurons.

We agree and changed the Introduction adding a description of EC neuroanatomy and role in slow oscillations. We also removed from the Introduction references to psychological processes, psychiatric disorders. We kept the description of the relationship between sleep/wake cycles and 5-HT as this provides a frame of reference to the relationship between 5-HT and slow oscillations. We address neuromodulation in the Discussion.

“Som interneurons are specifically known to be sufficient to cause desynchronization in V1 (Chen et al., 2015), while it is well known that they are potently excited by ACh in various cortical areas (Chen et al., 2015, Obermayer et al., 2018, Fanselow et al., 2008) including mEC (Desikan et al., 2018) and NE in frontal cortex (Kawaguchi and Kubota, 1998), our work identifies them as a novel target of 5-HT regulation via 5-HT_2A_R. Recently it has been proposed that inhibitory interneurons play a key role in mediating the effect of ACh and NE on cortical state transitions (Cardin, 2019), our results add a new level of complexity to this picture.”

Reviewer #3:The manuscript by De Filipo et al. presents a well-executed and novel study on the role of somatostatin positive (Som) interneurons in the generation of cortical slow oscillations (SO) in the entorhinal area. The authors employ a combination of mouse in vivo recordings under anesthesia, slice physiology, together with pharmacological and chemo/optogenetic manipulations to shown that entorhinal SO are suppressed by serotonin activation of Som interneurons. The present work provides new data on an interesting topic that can have broader implications to understand the relationship between neuromodulators, state-dependent brain oscillations and different interneuron subtypes. The experimental approaches are rigorous and adequately support the authors claims. However, the study has some limitations that cast doubts on the specificity of the effects reported. Additional analysis may be sufficient to clarify these doubts.1) The authors need to show more convincing evidence that the effects 5-HT and the manipulations reported in Figure 1 are specific of the SO and not a general dampened of entorhinal activity. More extensive analysis of the current data will suffice for that purpose. Figure 1H suggests that there may be a reduction in power in other frequencies than δ, but is not clearly appreciated (using a log frequency axis will facilitate it). A power spectrum comparison before, during and after drug application is necessary to evaluate the changes in all frequencies. Entorhinal cortex displays prominent γ oscillations, even during sleep and urethane anesthesia, are those also affected? This should be specifically quantified. Under urethane there is an alternation between a SO and a slow theta (~3-4Hz) state. Are theta oscillations also affected by the pharmacological manipulations?

A comprehensive analysis of the effect of MDMA/Fenf to different frequency bands has been added in Figure 1—figure supplement 2. Prominent theta oscillations were not observed in our experiments (Figure 1-2), at the start of every recording session an additional dose of urethane (0.2 g/kg) was injected to stabilize upstates frequency, in the following 30 min we never observed strong theta oscillations.

2) More comprehensive analysis of the effect of the different manipulations on SO reported through the manuscript should be performed, not only the rate of up states. For example, distribution of durations, inter-event intervals, amplitude of the remaining up states, etc. These will help to better understand the nature of the effects of serotonin release on SO.

Additional analysis has been performed in: Figure 1—figure supplement 3, Figure 3—figure supplement 1 and Figure 3—figure supplement 5.

3) Expanding the analysis and discussion of the contribution of PV interneurons will help establish the specificity of the role of Som interneurons. The results of the experiment with hM4Di-PV are encouraging in this regard. The authors do have fast-spiking, narrow-waveform cells (thus putative PV+ interneurons) in their dataset; however, there is very little discussion about this through the text. The results in Figure 4 provide evidence of the ability of Som interneurons to suppress SO. However, I would expect a very similar effect from PV activation. Unless the authors have evidence of the opposite, they should acknowledge this.

We agree with the reviewer that PV neurons activation have an effect on upstates that is comparable to Som neurons, we explicitly state this in the revised text citing a recent work that investigated effect of Som and PV neurons on upstates (Zucca et al., 2017). Our in vivo data (Figure 2) show a small number of FS units (TP latency < 0.4 ms) activated by 5-HT. However, in vitro, we did not see any activation from FS neurons (Figure 4—figure supplement 6). We acknowledge that the sample size of FS neurons is too small to conclusively talk about the role of PV neurons, moreover, PV neurons have been shown to be activated by 5-HT in prefrontal cortex (Athilingam et al., 2017, Puig et al., 2010). In conclusion, we do not exclude that a minority of FS cells are activated by 5-HT, even in entorhinal cortex, but in our opinion, this does not detract from the finding that Som neurons are activated by 5-HT, a discovery not previously reported in any brain area. Following the reviewer suggestion, we are open to perform additional analysis on PV neurons of our in vivo dataset.

Reviewer #4:In this manuscript, De Filippo et al. identify somatostatin-containing interneurons (SOM) of the entorhinal cortex (EC) as targets of serotonergic neurons in the dorsal raphe and describe their involvement in the mediation of slow oscillations in the EC, using an array of techniques such as pharmacology, optogenetics and DREADD.The manuscript is well and the analysis is solid. While there is no doubt this study demonstrates convincingly the involvement of SOM neurons in the regulation of SO, it fails to convincingly rule out other cell types which could equally contribute to the cortical dynamics under investigation. In that respect, I am left with some lingering doubts as to whether or not something was missed in the study and would therefore appreciate some clarifications on the part of the authors (see major point below).1) Some previous studies (for example, Ferezou et al., 2002) have shown that VIP/CCK GABAergic cells express 5-HT3 receptors, which are ionotropic receptors and can therefore respond very fast (with a depolarization, therefore an activation) to serotonergic neuromodulation. How come the authors do not mention this study (and others) and could they give an explanation as to why none of these neocortical cells (presumably also present in the EC?) were identified in their study as activated? Alternatively, using mCPBG in a similar fashion to how α-methyl5-HT was used in Figure 3 to rule out involvement of 5-HT3R would be valuable. The authors also mention the fact that only 11% of the cells expressing 5-HT2AR promoter were SOM. Could they please elaborate on the remaining 89% of cells?

We thank the reviewer for this concern. We have performed additional experiments to address the role of the 5HT_3_ expressing interneurons. Our results show that their involvement in the modulation of upstates in the medial entorhinal cortex is negligible (Figure 3—figure supplement 4). The 5HT_3_agonist, m-CPBG caused no reduction in the upstate frequency and the 5HT_3_ antagonist, Tropisetron failed to block the suppression of upstates by serotonin.

The majority of 5HT_2A_ cells were found in L6 as also reported by Weber and Andrade (2010). This is reflected by the peak density at ~800 µm in Figure 4—figure supplement 9 E (left). We did not analyze further the dataset using the 5-HT_2A_R-EGFP mouse (Weber and Andrade, 2010) as we discovered from personal communication with Dr. Andrade that in this mouse not all cells expressing 5-HT_2A_R are expressing the GFP marker. Future studies will be conducted using 5-HT_2A_R-cre mice.

2) Studies in the somatosensory cortex of awake mice (presumably therefore at a time when 5-HT release is highest) showed that L2/3 SOM neurons were not correlated with other neuronal cell types at the level of their membrane potentials and that they hyperpolarized and therefore reduce their spiking during active wake (when the mouse movedits whiskers (Gentet et al., 2012, Muñoz et al., 2017)). The authors should discuss their findings in the light of these papers considering that they argue that serotonin activates SOM neurons in the EC and 5-HT release is highest during wake, and therefore presumably, even more so during active wake.

We thank the reviewer for this question. We now discuss the relationship between sleep-wake and Som interneurons in the text:

“5-HT levels in the brain, similarly to other neuromodulators, are synchronized to the sleep wake cycle, with higher levels present during wake state (Oikonomou et al., 2019). How does this notion relate to activation of Som interneurons by 5-HT_2A_R? A limitation of our study is the absence of data in naturalistic conditions, with anesthesia potentially being a confounding factor (Adesnik et al., 2012). We can nonetheless speculate that Som interneurons should be more active during states with higher 5-HT levels (wake>SWS>REM). A previous study directly measuring the activity of various neurons classes across different states seem to support this idea. Som interneurons in the dorsal cortical surface display highest activity during wake, lower during SWS and lowest during REM (Niethard et al., 2016)… “

In a recent review from our lab (Tukker et al., 2020) we have discussed important differences between the somatosensory and entorhinal cortices in the initiation, propagation and regulation of up down states. Although we have not investigated Som neurons in the somatosensory cortex and their modulation by serotonin in this paper regarding up-down states, we predict that there might be differences compared to entorhinal cortex. Further in the paper from Muñoz et al., 2017 it is evident that Som has layer specific modulation / participation during whisker stimulation. Further studies will hopefully investigate the modulation of Som interneurons in a layer specific manner during brain states in different cortices.